# The *tae-miR164*-*TaNAC6A* Module from Winter Wheat Could Enhance Cold Tolerance in Transgenic *Arabidopsis thaliana*

**DOI:** 10.3390/plants14182849

**Published:** 2025-09-12

**Authors:** Ziyao Dai, Xiaoyan Yang, Wenwang Shan, Yiou Hao, Da Zhang, Kankan Peng, Qinghua Xu

**Affiliations:** College of Life Science, Northeast Agricultural University, Harbin 150030, China; daiziyao224776@163.com (Z.D.); yang20230223@163.com (X.Y.); shiq18279@gmail.com (W.S.); 17804620855@163.com (Y.H.); zhangda@neau.edu.cn (D.Z.); pengkankan@neau.edu.cn (K.P.)

**Keywords:** winter wheat, *tae-miR164-TaNAC6A* module, stress response mechanism, cold-stress tolerance

## Abstract

Cold stress impedes the growth and development of wheat (*Triticum aestivum*) and other crops, ultimately reducing both yields and quality. Research indicates that non-coding RNAs (ncRNAs) play a crucial role in regulating plant stress responses and resistance. In a previous study, we observed that the expression of *tae-miR164* was inversely correlated with the expression of *TaNAC6A* in Dongnongdongmai 1 (Dn1), a winter wheat variety with high cold resistance, under cold-stress conditions. However, the molecular mechanism governing the cold responsiveness of the *tae-miR164-TaNAC6A* module was not fully understood. We found *that tae-miR164* and *TaNAC6A* were both induced to express in opposite trends, and TaNAC6A was located in the nucleus. We also discovered that the expression of *tae-miR164* and its target gene, *TaNAC6A*, was responsive to short-term freezing stress in transgenic Arabidopsis plants. Compared to wild-type (WT) Arabidopsis plants, OE-*tae-miR164* plants showed decreased cold tolerance, whereas those overexpressing *TaNAC6A* demonstrated increased tolerance. On average, the OE-*TaNAC6A* and STTM-*tea-miR164* plants exhibited fewer morphological abnormalities in response to cold stress, higher antioxidant enzyme activities and gene expression levels, lower levels of reactive oxygen species (ROS) and malondialdehyde (MDA), and higher expressions of *AtDREB1*, *AtDREB2*, and *AtABI5* in the cold-signaling pathway. Thus, the biological functions of *tae-miR164* and *TaNAC6A* were initially confirmed through heterologous expression strategies, and we have made the first attempt to elucidate its associated *tae-miR164*-*TaNAC6A* module of cold resistance. The findings of this research will support further investigations into the regulation of plant stress resistance by ncRNAs and will inform molecular module breeding strategies aimed at enhancing the cold tolerance of crop plants. Molecular module design breeding, as a significant breakthrough in modern biotechnology, is transforming traditional breeding models. Conventional hybrid breeding relies on empirical screening, which is time-consuming and subject to randomness. In contrast, molecular module breeding directly targets key genes and achieves precise regulation through technologies such as gene editing and synthetic biology.

## 1. Introduction

Temperature is well-known to affect plant growth and development, particularly exposure at low temperatures [1]. Most plants are able to adapt to gradually decreasing temperatures by altering their cold-responsive gene expression to acclimate their physiology and biochemistry [2]. Frost damage occurs at critical temperatures when ice nucleation begins, resulting in structural and membrane damage. Freezing-induced increases in membrane permeability result in solute loss, ionic imbalance, and organelle damage [3]. It is therefore critical to understand the molecular mechanism underlying cold-stress resistance in order to develop cold-tolerant crops.

Adaptation to cold stress entails a complex array of physiological and biochemical mechanisms. In most plants, cold acclimation typically involves the synthesis and accumulation of proline (Pro) and other amino acids, proteins, sugars, and antioxidants [4]. Cold stress disrupts plant photosynthesis and leads to the accumulation of excessive ROS, resulting in lipid peroxidation, increased membrane permeability, cytoplasmic leakage, elevated relative conductivity, and inhibited development [5,6]. In response, plants express antioxidant enzymes, such as superoxide dismutase (SOD), catalase (CAT), ascorbate peroxidase (APX), and peroxidase (POD), to neutralize ROS [7]. For instance, Guo et al. [8] observed that, in both rice varieties LJ25 (low-temperature stress resistance) and LJ11 (low-temperature stress sensitivity), cold stress prompted the accumulation of Pro, soluble protein, glutathione (GSH), SOD, and APX. However, an opposite trend in ROS and malondialdehyde (MDA) was observed in both varieties. This may reflect differences in the antioxidant capacity and membrane lipid peroxidation levels between the two rice varieties when coping with low-temperature stress.

In plants, miRNAs have been identified as playing a core regulatory role in the response to cold stress [9]. The low-temperature responsiveness of miRNAs was first observed in Arabidopsis (*Arabidopsis thaliana*) [9], and it was subsequently reported in poplar (*Populus*) [10], rice (*Oryza sativa*) [11], wheat [12], tomato (*Solanum lycopersicum*) [13], grape (*Vitis vinifera*) [14], and cotton (*Gossypium hirsutum*) [15]. In tomato, the overexpression of *Sly-miR319d* enhances cold tolerance by inhibiting the expression of GAMYB-like1 [16]. In transgenic Arabidopsis plants, the overexpression of *tae-miR399* from wheat inhibited the expression of *AtUBC24*, which, in turn, inhibited the AtUBC24-mediated degradation of *AtICE1*, inducing the CBF signaling pathway, activating antioxidant defenses, and enhancing cold tolerance [17]. The highly conserved *miR164* has been found to regulate not only plant development but also defense responses to environmental and biological stress signals [18,19]. In tomato, *Sly-miR164A* regulated the fruit ripening and quality by modulating the expressions of *SlNAM2* and *SlNAM3*. *Sly-miR164A* was preferentially expressed during late fruit development, thereby regulating the fruit ripening and quality [20]. In wheat seedlings, the tae-*miR164-TaNAC14* module regulated root growth and development as well as stress (drought and salt) tolerance. Specifically, *TaNAC14* promoted root growth and development and enhanced drought tolerance, while *tae-miR164* inhibited root development and decreased drought and salt tolerance by downregulating the expression of *TaNAC14* [21].

NAC (NAM, ATAF, and CUC) transcription factors (TFs) are one of the largest TF families in plants and play important roles in their growth, development, and environmental stress response [22]. For instance, the overexpression of *CaNAC064* from pepper (*Piper nigrum*) in transgenic Arabidopsis plants has been shown to reduce the MDA content, cold-injury index, and relative electrolytic leakage while increasing the antioxidant activity in response to cold stress [23]. Similarly, the overexpression of *VvNAC17* from grape in transgenic Arabidopsis plants enhanced drought tolerance by upregulating the expressions of abscisic acid (ABA)- and stress-related genes [24]. Lastly, the overexpression of *ZmSNAC13* of maize (*Zea mays*) in transgenic Arabidopsis plants improved the tolerance to both drought and salt stresses. In this case, the primary stress tolerance mechanism was found to involve the ABA pathway and the MAPK signaling cascade [25].

Wheat is one of the most important cereal crops. Notably, Dongnongdongmai 1 (Dn1), a hexaploid (2n = 6x = 42, AABBDD) wheat, was reported to be the first winter wheat variety capable of safely overwintering in Heilongjiang, China, exhibiting a greening rate of more than 85% [26]. This remarkable attribute makes Dn1 a promising research candidate to understand the molecular mechanism of cold-stress tolerance in wheat. In a previous study, we observed that the expression of *tae-miR164* in Dn1 was negatively correlated with the expression of *TaNAC6A* (TraesCS5B02G054200, loc: chr5B (-) 59,165,237 -59170277) in the field (−25 °C). However, the regulatory mechanism underlying the *tae-miR164-TaNAC6A* module remained unclear. Therefore, we sought to reveal this molecular mechanism by characterizing the biological functions of the *tae-miR164*-*TaNAC6A* module in regulating cold resistance. This study clarifies the key molecular modules of cold resistance in Triticum aestivum and their mechanisms of action, promotes the exchange of superior cold-resistant genetic resources, facilitates the cultivation of cold-resistant crop varieties, breaks through breeding bottlenecks, and provides new impetus for the development of the *Triticum aestivum* seed industry.

## 2. Results

### 2.1. Analysis of Conserved Motifs and Gene Structures of tae-miR164 and TaNAC6A

Using RNAfold to predict the secondary structure, we found that pre-tae-miR164 exhibited a complex but stable stem–ring structure. According to WebLogo analysis, the mature miR164 sequence is highly conserved. In order to verify whether *tae-miR164* regulates the cold-stress response, several cis-acting elements within the pre-tae-miR164 promoter were identified using PlantCare, namely, LRE, Me-JAR, GARE, LTR, ABRE, and an enhancer (Appendix A).

TaNAC6A consists of 306 amino acids, with a relative molecular weight of 33.50 kDa. TaNAC6A contains a typical conserved NAM domain and is predicted to be localized to the nucleus. The *ProTaNAC6A* promoter contains a variety of cis-acting elements, including LTR, SARE, ABRE, and DRE, among others (Appendix A).

Bioinformatics predictions consistently indicate that the transcription of *tae-miR164* and *TaNAC6A* is induced by low temperature.

### 2.2. Expressions of tae-miR164 and TaNAC6A of Dn1 Under Cold Stress

RT-qPCR was conducted to assess the expression levels of *tae-miR164* and *TaNAC6A* in the tillering nodes and leaves of Dn1 under cold stress.

In tillering nodes, the *tae-miR164* expression peaked at 5 °C and was the lowest at −25 °C, decreasing with colder temperatures.

The *TaNAC6A* expression initially rose then fell, with a minimum at 5 °C and a maximum at −10 °C, showing a threefold increase. At −25 °C, the *TaNAC6A* expression was half that at −10 °C and 2.5 times higher than that at 5 °C (Figure 1A).

In leaves, the *tae-miR164* expression peaked at 0 °C, initially rising then falling. The TaNAC6A expression increased with lower temperatures, peaking at −25 °C (Figure 1B).

Both the *tae-miR164* and *TaNAC6A* levels were higher in leaves than in tillering nodes, with *tae-miR164* negatively regulating *TaNAC6A*.

This suggests that TaNAC6A’s elevated expression may relate to winter wheat cold resistance (Dn1).

### 2.3. Subcellular Localization of the TaNAC6A Protein

The pCAMBIA2300-*TaNAC6A-EGFP* fusion vector was developed for transient expression in tobacco to study the subcellular localization of TaNAC6A pr. A GFP signal was observed in the cell membranes, cytoplasm, and nuclei of tobacco leaves injected with *Agrobacterium tumefaciens* strain GV3101 carrying pCAMBIA2300-35S*-EGFP*. In contrast, green fluorescence was detected exclusively in the nuclei of tobacco leaves injected with *Agrobacterium tumefaciens* strain GV3101 carrying pCAMBIA2300-*TaNAC6A-EGFP* (Figure 2). This observation indicated that TaNAC6A is a protein located in the nucleus, and it may play a regulatory role.

### 2.4. Verification of Low-Temperature Responsiveness of Pre-tae-miR164 and TaNAC6A

Biogenic analysis predicted the presence of low-temperature-responsive elements within the promoters of *pre-tae-miR164* and *TaNAC6A*. These predictions were substantiated by examining the promoter activities of these two genes under low-temperature conditions. The promoters of *tae-miR164* and *TaNAC6A* were capable of inducing GUS expression at normal temperatures (Figure 3A,B). However, after 4 h of exposure to 4 °C, the GUS staining appeared lighter in the *PromiR164* tobacco leaves and more intense in the *ProTaNAC6A* tobacco leaves (Figure 3C,D). These outcomes indicate that *tae-miR164* and *TaNAC6A* can respond to low temperature (4 °C), which might trigger the expression of key genes in the cold-signaling pathway.

### 2.5. Validation of the tae-miR164-TaNAC6A Regulatory Relationship

To verify the relationship between *tae-miR164* and *TaNAC6A*, specific primers were designed at the 5′TUR of *TaNAC6A mRNA* (802 bp). PCR amplification using the 5′RACE technique yielded a single target band of 232 bp, corresponding to the size of the sequence after the first cleavage of *TaNAC6A* mRNA (Figure 4A). Sequencing of ten positive clones revealed that *tae-miR164* cleaved the target *mRNA* at a single site, located between the 10th and 11th bases from the 5′ UTR of *tae-miR164* (Figure 4B).

The relationship between *tae-miR164* and *TaNAC6A* was further investigated using transient co-transformation in tobacco. Four types of *Agrobacterium tumefaciens* strain GV3101 were prepared: the first with the pBI121-*GUS* construct, the second with the pBI121-*pre-tae-miR164* construct, the third with the pBI121-*TaNAC6A* construct, and the fourth consisting of a mixture containing both the pBI121-*pre-tae-miR164* and pBI121-*TaNAC6A* constructs. These types were individually infiltrated into tobacco leaves. The GUS phenotype observed in leaves inoculated with the pBI121-*GUS* empty vector (Figure 5A) was comparable to that seen in leaves treated with the pBI121-*TaNAC6A* strain (Figure 5C). Conversely, leaves infiltrated with the pBI121-*pre*-*tae-miR164* strain exhibited no GUS staining, as *pre-tae-miR164* suppressed the GUS gene expression (Figure 5B). Furthermore, leaves co-transformed with pBI121-*pre-tae*-*miR164* and pBI121-*TaNAC6A* displayed significantly reduced GUS-staining intensities (Figure 5D), suggesting that *tae-miR164* negatively regulates the expression of *TaNAC6A*.

The interaction between *tae-miR164* and *TaNAC6A* was validated once more using a dual-luciferase assay. Following the successful construction of the pGreenII-62-SK-*pre-tae-miR164* expression vector, the pGreenII 0800-tae-*miR164-TaNAC6A* and pGreenII 0800-*tae*-*miR164-*m*TaNAC6A* (with a mutation in the *TaNAC6A* cleavage site by *tae-miR164*) expression vectors were also successfully constructed. The constructed vectors were introduced into the *Agrobacterium tumefaciens* strain GV3101 and subsequently expressed in tobacco leaves. The fluorescence intensity was observed to determine whether a targeting relationship exists between *tae-miR164* and *TaNAC6A*. The fluorescence intensity was significantly reduced in tobacco leaves co-infiltrated with *Agrobacterium tumefaciens* strain GV3101 containing the SK-*tae-miR164* plasmid and the LUC-*TaNAC6A* plasmid. However, the fluorescence intensity was enhanced in tobacco leaves co-infiltrated with the SK-*tae-miR164* plasmid and the LUC-m*TaNAC6A* plasmid, which was introduced through an MRE (miRNA response element/miRNA binding site) mutation (Figure 6).

The three results collectively confirm that *tae-miR164* targets and negatively regulates the expression of *TaNAC6A*. And it was found that the 5′ base pairing is indeed essential for MRE to exert its crucial function. *tae-miR164* and *TaNAC6A* indeed exhibit an interaction relationship, and the *tae-miR164*-*TaNAC6A* module performed biological functions.

### 2.6. Phenotypic Changes in Transgenic Arabidopsis Plants

To investigate the functions of *tae-miR164* and *TaNAC6A* in relation to cold stress, we constructed overexpression vectors for 35S-*tae-miR164* and 35S-*TaNAC6A* and transformed them into Arabidopsis plants. From these transformations, we selected and obtained some OE-*tae-miR164* and OE-*TaNAC6A* plants; among them, the OE-*tae-miR164*-3, OE-*tae-miR164*-7, OE-*TaNAC6A*-4, and OE-*TaNAC6A*-8 plants were high-expression lines. Additionally, we constructed the STTM-*tae-miR164* (Short Targeted Tandem Motif knockout of *tae-miR164*) recombinant plasmid and transformed it into Arabidopsis plants. Subsequently, the third generation of genetically stable transgenic Arabidopsis plants, STTM-*tae-miR164* plants, were selected; among them, the STTM-*tae-miR164*-1 plants and STTM-*tae-miR164*-2 plants were the lines in which *tae-miR164* was most effectively knocked down. The *nac6* mutant (SALK K135734C, TARI) originated from the Columbia accession (Col-0), with the mutation site located in the homologous gene *AtNAC1* (AT1G56010) corresponding to *TaNAC6A* (Appendix A and Appendix A).

The phenotypes of five Arabidopsis plants (WT, OE-*tae-miR164*, OE-*TaNAC6A*, STTM-*tae-miR164, and nac6*) were observed under freezing stress (Figure 7A). No significant phenotypic differences were noted among the plants between 24 °C and 4 °C. However, at −10 °C, all plants displayed some degree of leaf curling or color deepening. Compared to the STTM-*tae-miR164* and OE-*TaNAC6A* plants, the WT, OE-*tae-miR164*, and *nac6* plants exhibited more pronounced color changes. Overall, the OE-*tae-miR164* plants appeared to be the most affected by low temperature. Following a 7-day recovery period, both the STTM-*tae-miR164* and OE-*TaNAC6A* plants returned to normal, whereas almost all the WT, OE-*tae-miR164*, and *nac6* plants perished. The survival rates of the STTM-*tae-miR164* and OE-*TaNAC6A* plants was about 80%, which were significantly higher than that of the WT, OE-*tae-miR164*, and *nac6* plants (Figure 7B).

Direct observation of the leaves from various lines revealed distinct phenotypic differences (Figure 8A). Compared to the WT, OE-*tae-miR164*, and *nac6* plants, the leaves of the STTM-*tae-miR164* and OE-*TaNAC6A* plants exhibited increased serrations along the leaf margins. Additionally, the leaves of the OE-*tae-miR164* plants displayed the lowest trichome densities. *nac6* also had relatively low trichome densities, although they were not significantly different from those of the WT. Moreover, the trichomes of the WT and OE-*tae-miR164* plants typically had two to three branches, whereas those of the STTM-*tae-miR164* and OE-*TaNAC6A* plants generally had three to four branches (Figure 8).

Under normal growth conditions, the primary roots of the OE-*TaNAC6A* plants were significantly longer than those of the WT and *nac6* plants. Furthermore, the primary roots of the STTM-*tae-miR164* plants were significantly longer than those of the WT and OE-*tae-miR164* plants. Finally, the primary roots of the OE-tae-miR164 plants were much shorter, and they had fewer lateral roots compared to the other four types of plants (Figure 9).

Many plants possess an outer epidermal layer composed of fatty acids or monohydric alcohols, known as the cuticle. This protective structure acts as a barrier against abiotic stressors, such as radiation, low temperature, and drought. In this study, we utilized scanning electron microscopy (SEM) to observe alterations in the cuticles of leaves cultivated under low temperature. At 24 °C, the OE-*TaNAC6A* and STTM-*tae-miR164* plants exhibited thicker cuticles than those of the WT, OE-*tae-miR164*, and *nac6* plants (Figure 10A). Following 3 days of exposure to 4 °C, the average cuticle thickness increased in all four transgenic Arabidopsis plants, mirroring the WT. Notably, the wax contents in the leaves of the OE-*TaNAC6A* and STTM-*tae-miR164* plants were significantly greater than those of the other three Arabidopsis plants (Figure 10B).

Stomata regulate the exchange of gases between the plant and the external environment. SEM observations indicated that the stomata of all the Arabidopsis plants were fully open at 24 °C. Following three days of exposure to 4 °C, the stomata of all lines tended to close to some degree. Notably, the stomata of the OE-*TaNAC6A* and STTM-*tae-miR164* plants exhibited some degree of openness as well as wax buildup (Figure 11).

The subcellular structures of four different Arabidopsis plants were observed using transmission electron microscopy (TEM). At 24 °C, the mesophyll cells of the four plants exhibited well-defined structures with clear organelle contours, with no significant differences observed. After 3 days of exposure to 4 °C, the mesophyll cells of the WT plants experienced severe plasmolysis and displayed abnormal nuclear and mitochondrial morphologies. Furthermore, their chloroplasts were deformed and disintegrating, with ruptured chloroplast membranes migrating towards the centers of the mesophyll cells. These deformities were markedly more severe in the OE-*tae-miR164* and *nac6* plants, exhibiting notable increases in both the numbers and sizes of starch granules. In contrast, the OE-*TaNAC6A* plants showed no significant changes, apart from a slight tendency for chloroplasts to move away from the edges of the mesophyll cells. Additionally, these plants had significantly reduced numbers of starch granules (Figure 12).

At 24 °C, the grana thylakoid structures in four plant genotypes were relatively well-ordered. Additionally, their mitochondria were round or oval and typically located either at junctions between chloroplasts and the plasma membrane or between two adjacent chloroplasts. After 3 days of exposure to 4 °C, the organelles in the WT plants began to disintegrate alongside vacuolization. Furthermore, the chloroplasts in the WT plants started to swell and deform, including the enlargement of starch granules. The number of grana decreased, and the thylakoid layers began to loosen and become disorganized, with some layers fusing together, resulting in blurred or missing thylakoid structures. However, in comparison to the WT plants, the OE-*TaNAC6A* plants retained normal chloroplast and mitochondrial structures even after being exposed to 4 °C for 3 days. This was evidenced by the absence of cell membrane disintegration and starch granule formation, indicating that the mesophyll cells could still maintain normal photosynthetic activity at this stage. In contrast, the cellular structures of the mesophyll cells in the OE-*tae-miR164* and *nac6* plants were significantly more damaged, with markedly increased numbers of starch granules. This suggested a substantial weakening in the ability to export photosynthetic products and an inability to perform normal photosynthesis. The numbers of starch granules increased in these plants, suggesting a weakened ability to export photosynthetic products (Figure 13).

### 2.7. Physiological Markers of Cold Resistance in Transgenic Arabidopsis Plants

The MDA content is one of the important indicators for evaluating the degree of peroxidation in the plant cell plasma membrane. The higher the MDA content, the greater the degree of membrane lipid peroxidation, and the more severe the cellular damage. Under normal conditions, plants contain relatively low Pro contents. However, when subjected to adverse environmental stresses, such as drought or cold, Pro accumulates in large amounts. The variation in conductivity can reflect the extent of cellular damage. Generally speaking, under cold stress, the lower the relative levels of electrical conductivity and MDA content, and the higher the Pro content, the less damage plant cells suffer, indicating stronger cold resistance in the plant.

Consequently, in order to evaluate whether the overexpression of *tae-miR164* and *TaNAC6A* enhances cold tolerance in transgenic Arabidopsis plants, the electronic conductivities and the contents of MDA and Pro were measured.

Overall, the electric conductivities of the five types of Arabidopsis plants tended to increase with decreasing temperature. At 4 °C and −10 °C, the electric conductivities of the STTM-*tae-miR164* and OE-*TaNAC6A* plants were significantly lower than those of the WT, OE-*tae-miR164*, and *nac6* plants (Figure 14A), indicating that the OE-*tae-miR164* plants suffered more severe damage from cold stress. Conversely, the cold resistance of the transgenic Arabidopsis plants was improved, in which the expression level of *TaNAC6A* was relatively high.

At 4 °C and −10 °C, the MDA contents in the Arabidopsis plants significantly increased compared to those at 24 °C, indicating that the cell membranes suffered varying degrees of oxidative damage. The MDA contents in the STTM-*tae-miR164* and OE-*TaNAC6A* plants were lower than those of the WT, OE-*tae-miR164*, and *nac6* plants at 24 °C, 4 °C, and −10 °C, although the difference was most pronounced at −10 °C (Figure 14B). Furthermore, the Pro contents in the STTM-*tae-miR164* and OE-*TaNAC6A* plants were higher than those of the WT, OE-*tae-miR1*64, and *nac6* plants at 4 °C and −10 °C (Figure 14C). It appears that the overexpression of *tae-miR164* reduced the cold tolerance of the *OE-tae-miR164* plants, whereas the overexpression of *TaNAC6A* enhanced the cold tolerance of the OE-*TaNAC6A* plants.

### 2.8. Cold-Induced ROS Production in Transgenic Arabidopsis Plants

Cold stress frequently leads to the overproduction of reactive oxygen species (ROS), including hydrogen peroxide (H_2_O_2_) and superoxide radicals (O_2_^·−^), among others. To ascertain whether the overexpression of *tae-miR164* and *TaNAC6A* influences the elimination of ROS, the levels of H_2_O_2_ and O_2_^·−^ were quantified in cold-stressed Arabidopsis plants. The DAB and NBT chemical staining assays indicated that the WT plants exhibited more intense staining than both the OE-*TaNAC6A* and STTM-*tae-miR164* plants at cold stress (4 °C and −10 °C). However, the OE-*tae-miR164* and *nac6* plants displayed darker staining than the WT plants (Figure 15A,B). The levels of H_2_O_2_ and O_2_^·−^ in the WT plants gradually increased as the temperature decreased, with significantly higher levels at cold stress compared to those at 24 °C. No significant differences were observed in the levels of H_2_O_2_ and O_2_^·−^ in the OE-*TaTaNAC6A* and STTM-*tae-miR164* plants at 4 °C compared to at 24 °C. However, the levels of H_2_O_2_ and O_2_^·−^ were higher at −10 °C than those at 24 °C. Furthermore, the levels of H_2_O_2_ and O_2_^·−^ were lower in both the OE-*TaTaNAC6A* and STTM-*tae-miR164* plants compared to those of the WT plants at both 4 °C and −10 °C. In contrast, the levels of H_2_O_2_ and O_2_^·−^ in the OE-*tae-miR164* and *nac6* plants were higher than those in the WT plants (Figure 15C,D).

### 2.9. Changes in Antioxidant Enzyme Activities and Gene Expression Levels in Transgenic Arabidopsis Plants Under Cold Stress

The SOD, POD, and CAT activities of the OE-*TaNAC6A* plants were the strongest under cold stress, followed by those of the STTM-*tae-miR164* and WT plants, while the *nac6* plants exhibited the lowest activity (Figure 16A,D,F). The RT-qPCR results for antioxidase-related genes (*AtSOD1*, *AtSOD2*, *AtPER3*, *AtCAT1*, *AtCAT2*, and *AtCAT3*) indicated that the expression levels of the OE-*TaNAC6A* and STTM-*tae-miR164* plants were consistently higher than those of the WT and *nac6* plants under the same low-temperature conditions (Figure 16B,C,E,G–I).

### 2.10. Alterations in the Expression of Cold-Signaling-Related Genes in OE-TaNAC6A Plants

Cold signaling is segregated into two major signaling pathways: ABA-independent and ABA-dependent pathways. In Arabidopsis, cold stress has been shown to induce the expression of ICE1, activating downstream DREB1/CBF members, which, in turn, induces an ABA-independent signaling pathway [27]. It is well-known that ABIs are key members in the ABA-dependent signaling pathway.

Therefore, we detected two genes (*AtDREB1*, *AtDREB2*) in the ABA-independent signaling pathway and three genes (*AtABI3, AtABI4*, and *AtABI5*) in the ABA-dependent signaling pathway by RT-qPCR. The results indicated that, as the temperature decreased, the expression levels of *AtDREB1*, *AtDREB2*, and *AtABI5* in the OE-*TaNAC6A* plants with high *TaNAC6A* expression exhibited a gradual increase. At 4 °C, the expression levels of *AtABI3* and *AtABI4* were both higher than those at 24 °C and −10 °C (Figure 17).

These results indicate that the overexpression of *TaNAC6A* enhanced the cold-signaling pathways in the Arabidopsis plants.

## 3. Discussion

An increasing body of evidence indicates that miRNAs are crucial in regulating plant growth, development, and stress tolerance [28]. In a previous study, high-throughput RNA sequencing libraries were constructed using tillering nodes from Dn1 as samples, collected at temperatures of 5 °C and −10 °C. Thirty miRNA families were identified in the library from samples at 5 °C, while 24 miRNA families were identified in the library from samples at −10 °C. A total of 53 genes were predicted to be targets of the 30 miRNA families in the 5 °C library [29]. To fully leverage Dn1’s abundant genetic resources for cold resistance, it is essential to characterize the functions and regulatory mechanisms of the miRNAs associated with cold resistance. miR164 is one of the most conserved miRNAs and is involved in plant growth, development, and abiotic and biotic stress responses [21]. One miRNA molecule can control multiple target genes. Spychała et al. found that *tae-miR164* targeted *Lr46-RLK3*, a rapid responsive gene to *Puccinia triticina* infection [30]. In Wang et al.’s research, *TaMAPK4* was verified to be a target of *tae-miR164*, which played an important role in signaling during the wheat–*Pst* interaction [31].

Previous studies have also shown that *tae-miR164* targets plant-specific NAC transcription factors from the NAM subfamily of wheat. Feng et al. revealed that *TaNAC21/22,* a NAC transcription factor gene, was the target of *tae-miR164* and played an important role in regulating the resistance of host plants to stripe rust [32].

Specifically, *miR164* negatively regulats the expression of its target NAC transcription factor (TF). We identified *TaNAC6A mRNA* as the target gene of *tae-miR164* through psRNATarget analysis. Subsequently, this study confirmed that the *tae-miR164-TaNAC6A* module enhanced the cold resistance of Dn1, following a combination analysis of 5′RACE, tobacco transient co-transfection, and dual-luciferase assays.

Research into the stress-regulating functions of NAC transcription factors (TFs) has primarily focused on drought and salt stresses, with limited studies exploring signal transduction under low-temperature stress, particularly at sub-zero temperatures. Zhou et al. identified that the transient overexpression of *TaNAC6A/6B/6D* reduced the haustorium index of Yangmai158, and the stable transformation of *TaNAC6A* enhanced its resistance against *Bgt*, implying that TaNAC6s play important roles in basal resistance [33]. *AtNAC1* in Arabidopsis is the homologous gene of *TaNAC6A* in wheat, which was analysized by WheatOmics. Rodríguez-García et al. identified that AtNAC1 activates the expression of peptidase-encoding *AtCEPs* in roots to limit the root hair growth of Arabidopsis plants [34]. Xie et al.’s study provided mechanistic insights into how AtNAC1 regulates root ground tissue maturation by coordinating with the SCR/SHR-CYCD6;1 module in Arabidopsis plants [35]. Xie et al. even discovered the AtNAC1 transduced auxin signal downstream of TIR1 to promote lateral root development [36]. Our study is the first to validate the biological functions of *TaNAC6A* in wheat and *AtNAC1* in Arabidopsis plants in regulating plant cold tolerance.

Dong et al. discovered, in their research on tomato, that the *miR164a-NAM3* module induces ethylene synthesis by directly regulating the expressions of *SlACS1A*, *SlACS1B*, *SlAC01*, and *SlAC04*, thereby conferring cold tolerance to transgenic tomato plants [37].

Li et al. treated wheat with phenanthrene (polycyclic aromatic hydrocarbons, PAHs) and found that phenanthrene stress (ubiquitous organic pollutants in the environment) accelerated the senescence and death of wheat roots while stimulating the emergence of new roots. Phenanthrene stress upregulated the expression of *tae-miR164* and enhanced the silencing of *TaNAC1*, thereby inhibiting the occurrence of adventitious roots [38]. Our study found that the *tae-miR164-TaNAC6A* module can regulate cold tolerance in transgenic Arabidopsis plants, and we validated this at the phenotypic, physiological, and molecular levels.

Following recovery from exposure to −10 °C, the survival rates for the OE-*TaNAC6A* and STTM-*tae-miR164* plants were 84% and 77.6%, respectively. In contrast, the survival rates for the OE-*tae-miR164*, *nac6*, and WT plants were 22%, 28.4%, and 34%, respectively. The OE-*TaNAC6A* plants exhibited a higher survival rate than the STTM-*tae-miR164* plants, indicating significant cold tolerance. Additionally, under cold stress, the OE-*tae-miR164* plants showed higher MDA contents and electrical conductivities and lower Pro contents compared to the other four lines.

In general, extremely low temperatures often result in the excessive production of ROS, the accumulation of MDA, and increased conductivity—biomarkers indicative of cell membrane damage [39]. Plants strictly regulate their ROS levels by recruiting enzymatic and non-enzymatic antioxidants [40,41]. The activities of SOD, CAT, and POD in wheat leaves increased after the Methyl jasmonate (MJ) pretreatment of the plants subjected to hardening temperature, as compared to the control [42]. In our study, at 4 °C and −10 °C, the leaves of the OE-*tae-miR164* and *nac6* plants exhibited deeper staining by DAB and NBT than those of the other three lines. Moreover, the H_2_O_2_ and O_2_^·−^ contents of the OE-*tae-miR164* and *nac6* plants increased with decreasing temperature. In addition, markers of membrane permeability and membrane lipid peroxidation increased in the OE-*tae-miR164* plants but decreased in the OE-*TaNAC6A* plants. The activities of SOD, POD, and CAT gradually decreased with decreasing temperature in the OE-*tae-miR164* and *nac6* plants. Finally, the expression levels of *AtSOD1/2*, *AtPER3*, and *AtCAT1/2/3* were lower in the OE-*tae-miR164* and *nac6* plants than those in the other lines, consistent with the antioxidant activity assays.

DREB TFs regulate plant stress resistance. DREB1 and DREB2 regulate two different signaling pathways to resist stress caused by cold or dehydration [43]. ABA plays a crucial role in regulating plant growth during stress responses. By using molecular biology approaches, significant positive regulatory genes associated with ABA responses have been identified, including *ABI3*, *ABI4*, and *ABI5* [44]. Here, we evaluated the differential expressions of five cold-stress-related genes in transgenic OE-*TaNAC6A* plants. The expression levels of *AtDREB1/2* and *AtABI3/4/5* were significantly altered by exposure to cold temperatures. Together, the results suggest that *tae-miR164* negatively regulates cold tolerance while *TaNAC6A* promotes cold resistance.

The leaf morphology, size, and number are crucial for plant growth and development [45]. Compared to the WT plants, the *OE-tae-miR164* plants exhibited smoother leaf margins, whereas the STTM-*tae-miR164* plants showed increased serration. Notably, the trichome density in the OE-*tae-miR164* plants was lower than that in the other lines. Most plant leaves possess a waxy cuticle, composed of fatty acids or monoalcohols synthesized by their epidermal cells [46], which serves as a barrier against abiotic stressors such as radiation, low temperatures, and drought. SEM observations revealed that both the OE-*TaNAC6A* and STTM-*tae-miR164* plants had thicker, more waxy cuticles, even under normal temperature conditions. After exposure to 4 °C for three days, the stomata of all five lines tended to close, and their cuticles thickened. Nonetheless, the cuticles of the OE-*TaNAC6A* and STTM-*tae-miR164* plants remained thicker than those of the other lines.

Exposure to low temperatures often results in plasma vesiculation, enlargement of chloroplasts and mitochondria, and the blurring or disappearance of nuclear and plasma membranes, which can sometimes lead to complete cellular disintegration [47]. In this study, TEM observations showed normal cell structures in all five lines at room temperature, with the OE-*TaNC6A* plants having larger starch granules. After three days of exposure to 4 °C, the WT, OE-*tae-miR164*, and *nac6* plants displayed fusiform to circular chloroplast swelling, internal vacuolation, increases in osmophilic bodies, and expanded and/or disintegrated starch granules. In contrast, the OE-*TaNC6A* plants exhibited a relatively stable cell structure. Overall, the OE-*tae-miR164* plants showed poor organellar stability in response to cold stress. Conversely, plants overexpressing *TaNC6A* demonstrated stable cellular structures in response to low temperature, indicating improved cold tolerance.

Given that wheat is the allohexaploid progenitor, it is well-known that its genome is vast and complex, resulting in slow progress in gene cloning and functional studies. Therefore, this study selected Arabidopsis plants for heterologous validation, which also was the potential limitation to the in-depth analysis of the molecular mechanism of the *tae-miR164-TaNAC6A* module in wheat.

## 4. Materials and Methods

### 4.1. Plant Materials, Growth Conditions, and Treatments

Seeds of winter wheat, Dn1 (with strong cold resistance), provided by our laboratory, were sown on 12 September 2020. Approximately 200 seeds were planted per row at 0.2 m intervals and a depth of 5 cm within a 2 m planting area. The seeds were covered with soil to promote natural germination, and the plants were managed using conventional practices. Under natural cooling conditions in the field, tillering nodes and leaves were sampled when the ten-day-average minimum temperature reached about 5 °C (on 10 October 2020), 0 °C (on 29 October 2020), −10 °C (on 25 November 2020), and −25 °C (on 3 January 2021) [48].

Arabidopsis seeds, ecotype Col-0, as wild type (WT), provided by our laboratory, Arabidopsis mutant seeds *nac6* (SALK135734C, TAIR), purchased from The Arabidopsis Information Resource (TAIR), and the other three transgenic Arabidopsis seeds constructed by this study were cultivated under a light intensity of 120–150 μmolm^−2^ s^−1^, 24 °C, a 16/8 h day/night photoperiod, and a relative humidity of 60%. After 28 days of cultivating, the Arabidopsis seedlings were cold-acclimated at 4 °C for 3 days and subsequently exposed to −10 °C for 3 h. Leaves were sampled at three distinct time points: before cold acclimation, after 3 days of cold acclimation, and after 3 h of exposure to −10 °C. Samples were wrapped in tin foil, immediately frozen in liquid nitrogen, and stored at −80 °C for future use [49]. Following the cold treatment, plants were grown at room temperature for an additional 7 days to calculate the survival rate. Plants that were dry and yellow and unable to regenerate were considered dead.

### 4.2. Gene Expression Analysis

Frozen tissues of winter wheat and fresh leaves of Arabidopsis plants were homogenized using liquid nitrogen. Total RNA was isolated from the plants using the Ultrapure RNA Kit (CW0581M, CWBIO, Suzhou, China). The cDNA was then synthesized from the mRNA with the HiScript III 1st Strand cDNA Synthesis Kit (R312, Vazyme, Beijing, China). RT-qPCR was employed to quantify the gene expression, following our previously published method [50]. The mRNA levels of the target genes were quantified with the 2^−∆∆CT^ method [51]. (Relevant RT-qPCR primers are listed in Appendix A).

### 4.3. Bioinformatics Analysis

The amino acid sequence of TaNAC6A, along with the gene and promoter sequences of *tae-miR164* and *TaNAC6A,* were obtained from wheatOmics (http://202.194.139.32/, 20 December 2024) The precursor and mature sequences of *tae-miR164* were sourced from miRBase (https://www.mirbase.org/, 20 December 2024) and Dn1 miRNA libraries previously established by our lab. The base conservation in the mature miRNA was analyzed using WebLogo (http://weblogo.berkeley.edu/, accessed on 8 September 2025).

RNAfold (http://rna.tbi.univie.ac.at/cgi-bin/RNAWebSuite/RNAfold.cgi, 12 January 2025) was used to predict the stem–loop structure of the tae-miR164 precursor. ExPASy (http://expasy.org, 12 January 2025) was utilized to determine the physical and chemical properties of TaNAC6A, including the amino acid length, molecular weight, isoelectric point, and aliphatic index. PlantCare (http://bioinformatics.psb.ugent.be/webtools/plantcare/html, 12 January 2025) was employed to obtain the pre-*tae-miR164* and *TaNAC6A* promoter sequences for cis element analysis. The subcellular localization prediction website (http://www.csbio.sjtu.edu.cn/bioinf/Cell-PLoc-2, 12 January 2025) was used to forecast the subcellular localization of TaNAC6A.

### 4.4. Subcellular Localization Analysis

The coding sequence (CDS) of *TaNAC6A* was constructed into the pCAMBIA2300 -EGFP vector. Tobacco leaves (seeded 28 days prior) were injected with *Agrobacterium tumefaciens* strain GV3101 carrying pCAMBIA2300-*EGFP* and pCAMBIA2300-*TaNAC6A-EGFP*. Seventy-two hours after the injection, the leaves were harvested to observe the EGFP signals under a fluorescence microscope (SZX10, Olympus, Tokyo, Japan).

### 4.5. Transient Expression Assay in Tobacco

To analyze the promoter activity of pre-*tae-miR164* and *TaNAC6A* and verify the regulatory relationship between *tae-miR164* and its target gene (*TaNAC6A*), a transient expression assay was performed on tobacco (*Nicotiana tabacum*). The pre-*tae-miR164* and *TaNAC6A* promoter sequences (2000 bp) were constructed into the pBI121-GUS vector (relevant PCR primers are listed in Appendix A). Tobacco leaves (seeded 28 days prior) were subsequently injected with *Agrobacterium tumefaciens* strain GV3101 carrying pBI121-*pre-tae-miR164-GUS* and pBI121-*ProTaNAC6A-GUS*. Seventy-two hours after the injection, the tobacco plants were exposed to 4 h at 4 °C. Subsequently, the leaves were harvested for histochemical staining.

### 4.6. 5′RACE Assay

The 5′RACE assay was conducted using a FirstChoice RLM-RACE Kit (Am1700, Thermo Fisher, Waltham, MA, USA), following the manufacturer’s instructions. (The 5′ primers, including 5′ RACE inner, and 5′ RACE outer, were provided by the kit; the 3′ primers are listed in Appendix A).

### 4.7. Dual-Luciferase Assay

The *pre-tae-miR164* sequence was constructed into the pGreenII-62-SK vector, and the upstream and downstream 100 bp sequences of the *tae-miR164* recognition site mutation of *TaNAC6A* were constructed into the pGreenII0800-miRNA vector. (Relevant PCR primers are listed in Appendix A.) Tobacco leaves, aged 28 days, were subsequently injected with *Agrobacterium tumefaciens* strain GV3101 (pSoup), carrying both control and target vectors at a 1:1 (*v:v*) ratio. The tobacco plants were subsequently cultured in the dark for 1 day and in the light for 2 days. The abaxial surfaces of the leaves, facing upwards, were sprayed with a reaction solution containing 1 mM of D-luciferin and then placed in a dark environment for 15 min. Following treatment, the leaves were observed using a chemiluminescence/fluorescence imaging analysis system (5200Multi, Tanon, Shanghai, China) to detect the luminescence.

### 4.8. Overexpression Vector Construction and Arabidopsis Transformation

The ORFs of *TaNAC6A*, *pre-tae-miR164,* and STTM-*tae-miR164* were separately cloned into pCAMBIA2300, which was driven by the CaMV35S promoter and then transformed into the *Agrobacterium tumefaciens* strain GV3101. Particularly, the sequence was constructed using the mature *tae-miR164* sequence as a template, with three bases (CTA) inserted between complementary bases at positions 9 and 10 at the 5′ end of *tae-miR164*. STTM-*tae-miR164* forms a stem–loop structure following transcription, and the complementary portions of *tae-miR164* and STTM-*tae-miR164* exist in the form of single strands, binding complementarily with *tae-miR164*. The *Agrobacterium tumefaciens* strain GV3101 containing the target vector was individually transferred into Arabidopsis seedling ecotype Col-0 via the floral dip method. The positive transgenic lines were first screened on MS plates containing 50 mg·mL^−1^ kanamycin, and then the seedlings were detected by PCR.

### 4.9. Phenotyping and Physiological Index Assays

The seeds of Arabidopsis plants were sown on 1/2 MS solid medium, stratified for 2 days at 4 °C, and transferred to growth chambers (16 h day/8 h night at 24 °C). The length of the main root was measured by Image J 1.47 software. After 28-day-old Arabidopsis seedlings were cold-acclimated at 4 °C for 3 days, the morphologies of the leaves, stomata, and cellular ultrastructure were observed by a stereomicroscope (SZX10, Olympus, Tokyo, Japan), a scanning electron microscope (SU8010, Hitachi, Tokyo, Japan), and a transmission electron microscope (HT7800, Hitachi, Tokyo, Japan), respectively.

Leaf samples were collected after treatment. The electrical conductivity (relatively) was measured by a conductivity meter (DDS-307A, Leici, Shanghai, China). The relative H_2_O_2_ contents of the leaves were assessed by DAB staining, and the relative O_2_^·−^ contents were assessed by NBT staining.

The contents of H_2_O_2,_ O_2_^·−^, Pro, and MDA were measured using one-to-one corresponding commercial kits from Comin Biotechnology (H_2_O_2_-2-Y, SA-2-G, PRO-2-Y, MDA-2-Y, Coming, Suzou, China), following the manufacturer’s instructions. By using a visible spectrophotometer (721, Jinghua, Shanghai, China), the absorbance at 415 nm, 530 nm, 520 nm, and 532 nm was measured to calculate the contents of H_2_O_2_, O_2_^·−^, Pro, and MDA.

The activities of SOD, POD, and CAT were measured using the relevant commercial kits from Comin Biotechnology (SOD-2-W, POD-2-Y, CAT-2-W, Comin, Suzhou, China), following the manufacturer’s instructions. For the SOD, POD, or CAT activity determination, 0.1 g leaf samples was homogenized in 1 mL of PBS extraction buffer on ice. The absorbance at 560 nm, 470 nm, and 240 nm was measured to calculate the activities of SOD, POD, and CAT using a visible spectrophotometer (721, Jinghua, Shanghai, China). All experimental data were calculated according to the fresh weight.

### 4.10. Statistical Analysis

Three biological replicates were used for each experiment. Data are presented as the means of three biological replicates ± standard deviations (SDs), and data analyses were performed using IBM SPSS 26. All data were subjected to two-way analysis of variance (ANOVA). GraphPad Prism 8.0 was used for in-depth data analysis and for generating figures. A value of *p* < 0.05 was used to determine significance.

## 5. Conclusions

Taken together, our study suggests that the overexpression of *tae-miR164* significantly downregulated the expression of *TaNAC6A* and reduced the cold tolerance of Arabidopsis plants. We identified that tae-miR164 and TaNAC6A might be involved in the low-temperature response. The subcellular location indicated that it might function as a transcription factor in cold-stress tolerance. We confirmed a physical interaction between *tae-miR164* and *TaNAC6A*. Meanwhile, the results of the *TaNAC6A* and *tae-miR164* overexpression in Arabidopsis plants indicated that the *tae-miR164*-*TaNAC6A* module played a key role in regulating the cold tolerance in the transgenic Arabidopsis plants in the epidermal hair development and leaf margin phenotype and in regulating the balance of ROS, as well as in promoting the expressions of *AtDREB1*, *AtDREB2*, and *AtABI5*. Conversely, the overexpression of *TaNAC6A* enhances the cold tolerance of Arabidopsis plants. Our research findings reveal the physiological and molecular mechanisms of the *miR164-TaNAC6A* module in winter wheat responding to cold stress, providing new insights for in-depth studies on the ncRNA regulation of plant stress resistance. This has significant guiding implications for the molecular module breeding of cold resistance in other crops in order to create crop varieties that better meet the needs of humans. But bottlenecks still exist in molecular module breeding technology, such as high costs and an incomplete understanding of the interaction mechanisms between module molecules.

## Figures and Tables

**Figure 1 plants-14-02849-f001:**
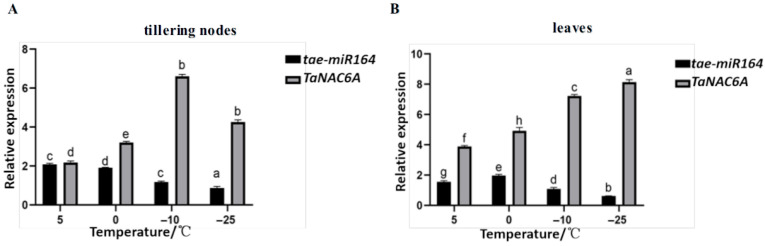
Expressions of *tae-miR164* and *TaNAC6A* in response to cold stress. (**A**) Expressions of *tae-miR164* and *TaNAC6A* in tillering nodes of Dn1. (**B**) Expressions of *tae-miR164* and *TaNAC6A* in leaves of Dn1. Values represent the means ± SDs (n = 10). Different lowercase letters indicate significant differences between treatments (*p* < 0.05), determined by two-way ANOVA. The reference gene used is *TaActin*.

**Figure 2 plants-14-02849-f002:**
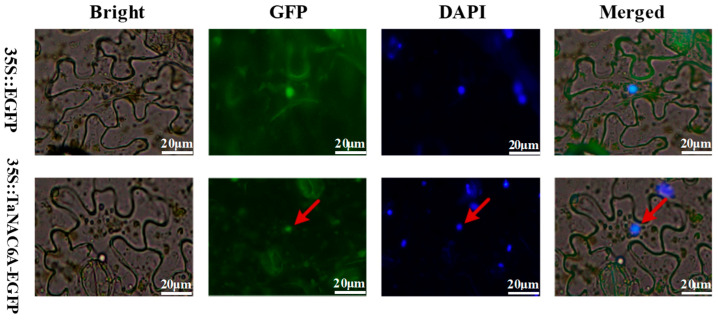
Subcellular localization of TaNAC6A protein. The Agrobacteria carrying 35 S:EGFP (control) and 35 S:*TaNAC6A-EGFP* vectors were transformed into the leaves of tobacco. The GFP signals of TaNAC6A-EGFP (**bottom**) and EGFP alone (**top**) were observed using a laser confocal microscope. DAPI was used as a nuclear marker in this experiment. Red arrow indicates the position of TaNAC6A-EGFP. Scale bars = 20 μm.

**Figure 3 plants-14-02849-f003:**
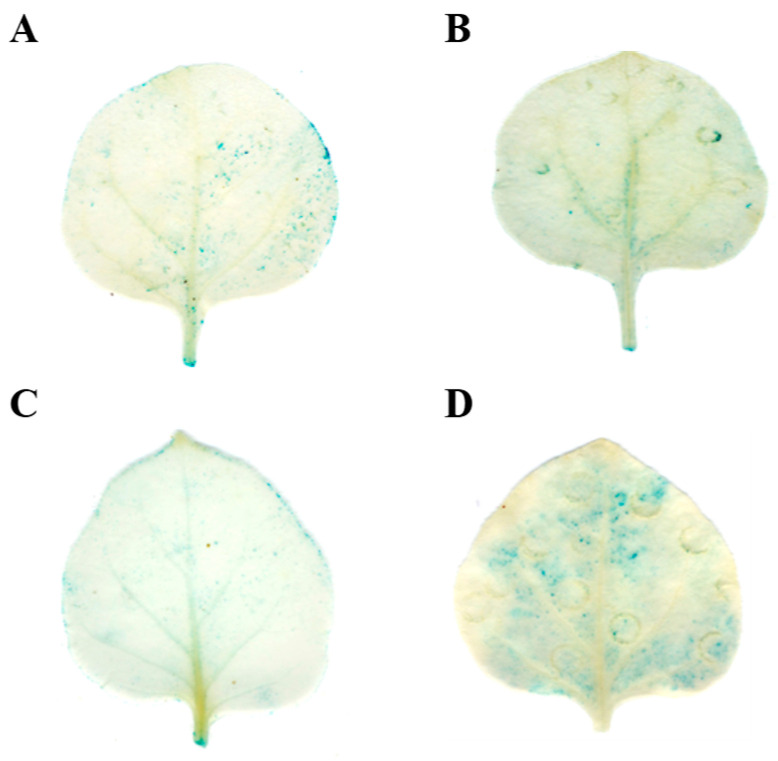
Analysis of promoter activity at different temperatures. GUS phenotype observed via histochemical staining. Leaves of tobacco separately injected with (**A**) 35S:*Pro-miR164-GUS* vector; (**B**) 35S:*Pro-TaNAC6A-GUS* vector; (**C**) 35S:*Pro-miR164-GUS* vector; (**D**) 35S:*Pro-TaNAC6A-GUS* vector after 4 h of exposure to 4 °C.

**Figure 4 plants-14-02849-f004:**
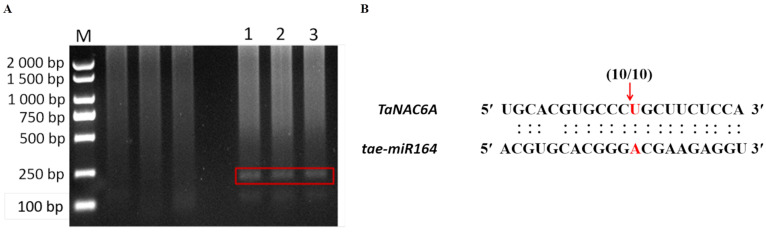
Interaction between *tae-miR164* and its target (*TaNAC6A*). (**A**) PCR amplification products after 5′RACE; red box indicates the amplified band. M: Marker 2000. Lanes 1, 2, and 3: cleaved *TaNAC6A* fragments; (**B**) model of the cDNA structure and mRNA cleavage site of *TaNAC6A*, as determined by 5′ RACE. Red letters show tae-miR164 complementary sites with the nucleotide positions of *TaNAC6A*. The expanded regions show the RNA sequence of each complementary site from 5′ to 3′ and the miRNA sequence from 3′ to 5′. The red arrow indicates a cleavage site verified by 5′-RACE, with the frequency of cloned PCR products shown above the alignment.

**Figure 5 plants-14-02849-f005:**
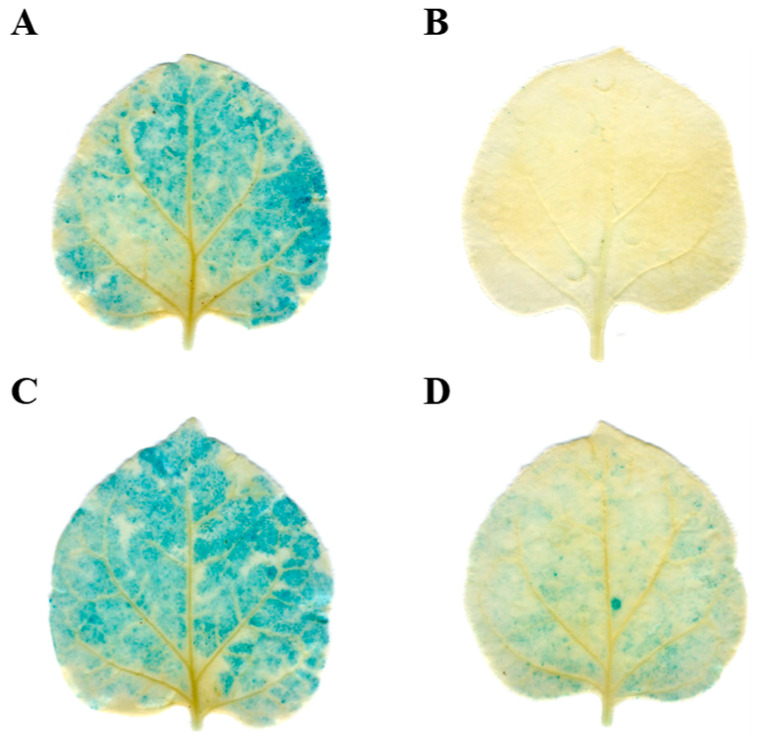
GUS phenotypes observed through histochemical staining. (**A**) Leaf injected with pBI121-*GUS* empty vector; (**B**) leaf injected with pBI121-*pre*-*tae-miR164* vector; (**C**) leaf injected with pBI121-*TaNAC6A* vector; (**D**) leaf injected with the mixture of pBI121-*pre-tae-miR164* and pBI121-*TaNAC6A* vectors.

**Figure 6 plants-14-02849-f006:**
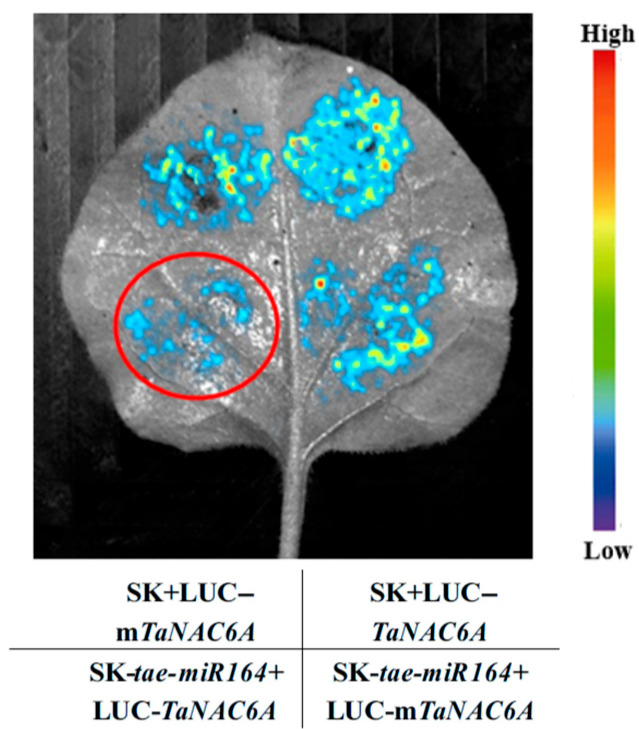
Dual-luciferase assay to verify the relationship between *tae-miR164* and *TaNAC6A*. Dual-luciferase assay showing the activation of SK-driven luciferase reporter gene in tobacco leaves. *Agrobacterium tumefaciens* strain GV3101 carrying different combinations of constructs was infiltrated into different regions of a tobacco leaf. The portion marked with a red circle shows that the luciferase activity was significantly inhibited. Images of luciferase activities were taken three days after the infiltration.

**Figure 7 plants-14-02849-f007:**
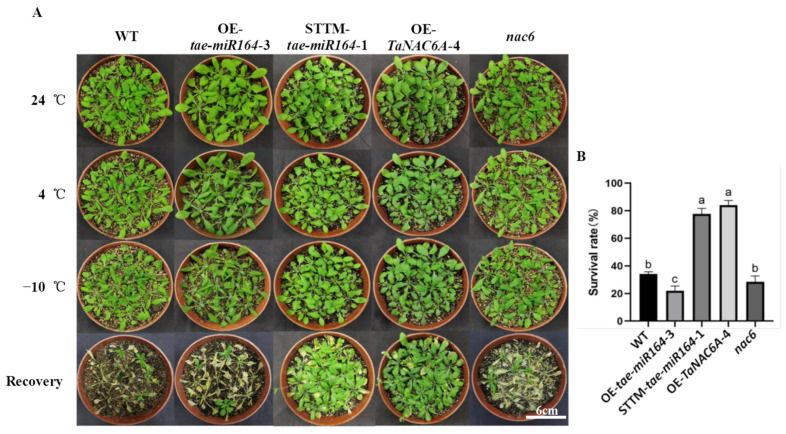
Phenotypes and survival rates of Arabidopsis plants under freezing stress. (**A**) Plant phenotypes; scale bars = 6 cm; (**B**) survival rates. Values represent the means ± SEs (*n* = 30). Different lowercase letters indicate significant differences between treatments (*p* < 0.05), determined by two-way ANOVA.

**Figure 8 plants-14-02849-f008:**
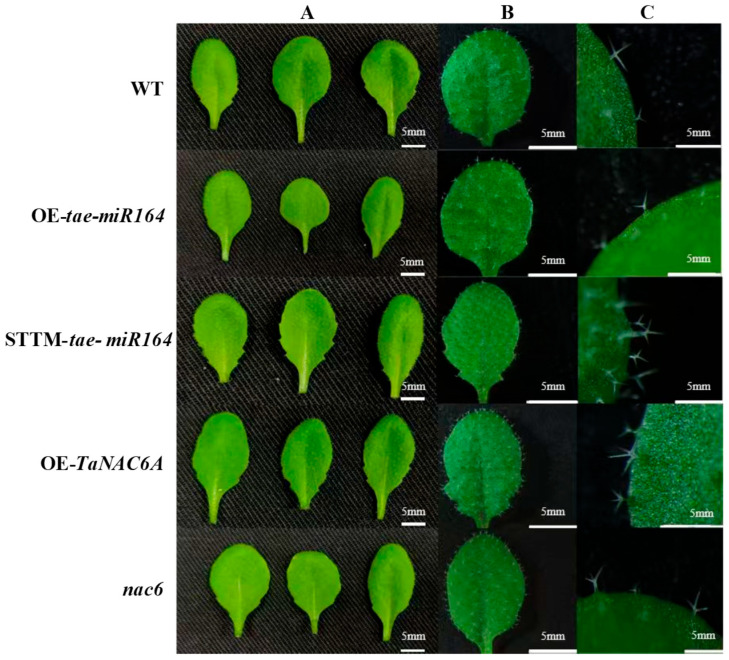
Changes in the leaf margins and trichomes of Arabidopsis plants. (**A**) Images of leaf margins from various lines; (**B**) increased trichome densities and (**C**) altered numbers of branches were associated with *tae-miR164* deletions and *TaNAC6A* overexpression. Images were taken from the third leaves of 14-day-old seedlings. OE-*tae-miR164* represents OE-*tae-miR164*-3 plants, STTM-*tae-miR164* represents STTM-*tae-miR164*-1 plants, and OE-*TaNAC6A* represents OE-*TaNAC6A*-4 plants. Scale bars = 5 mm.

**Figure 9 plants-14-02849-f009:**
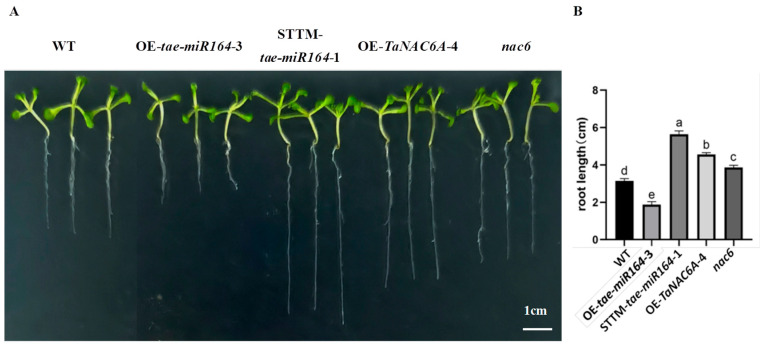
Root changes in Arabidopsis plants. (**A**) Lengths of primary roots at 10 days; scale bars = 1 cm; (**B**) statistical analysis of the lengths of the primary roots. Values represent the means ± SDs (n = 15). Different lowercase letters indicate significant differences between treatments (*p* < 0.05), determined by two-way ANOVA.

**Figure 10 plants-14-02849-f010:**
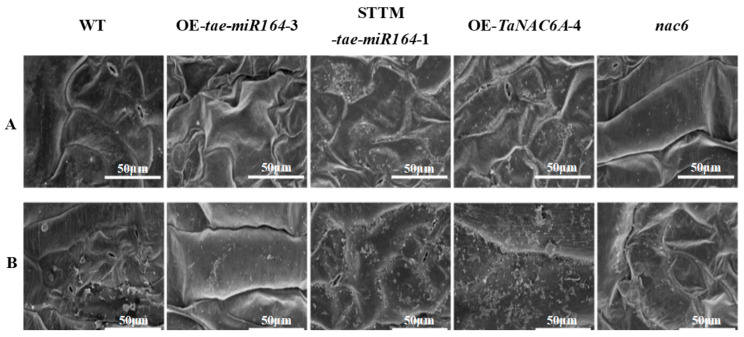
SEM micrographs of alterations in the cuticles of leaves of Arabidopsis plants. (**A**) OE-*TaNAC6A* and STTM-*tae-miR164* plants exhibited thicker cuticles at 24 °C; (**B**) wax contents of OE-*TaNAC6A* and STTM-*tae-miR164* plants were greater after 3 days of 4 °C cold stress. Scale bars = 50 μm.

**Figure 11 plants-14-02849-f011:**
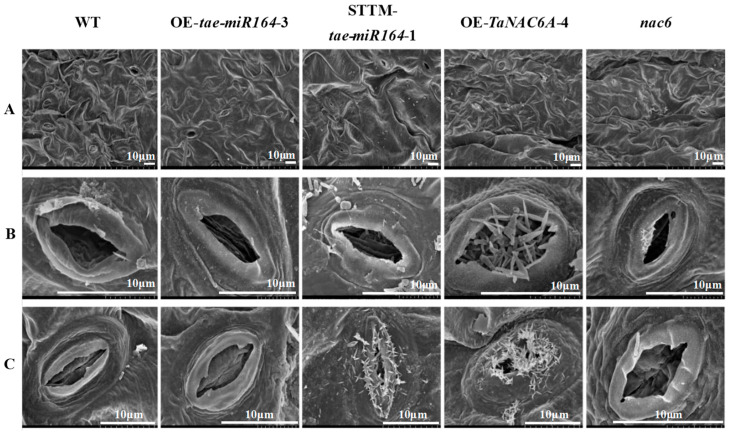
SEM micrographs of stomatal morphology changes in Arabidopsis plants. (**A**,**B**) Stomata of all Arabidopsis plants were fully open at 24 °C; (**C**) stomata of OE-*TaNAC6A* and STTM-*tae-miR164* plants still exhibited some degree of openness as well as wax buildup after 3 days of 4 °C cold stress. Scale bars = 10 μm.

**Figure 12 plants-14-02849-f012:**
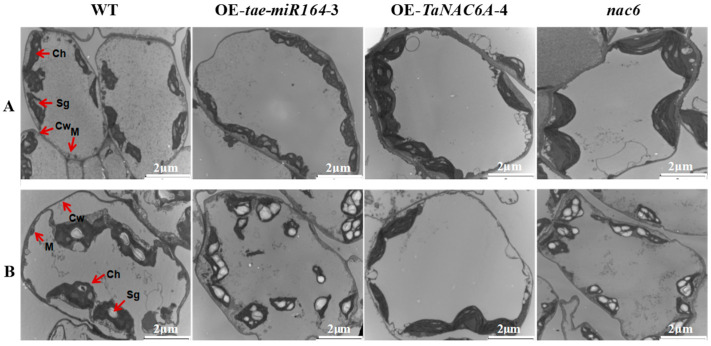
TEM micrographs of subcellular structure changes in mesophyll cells of Arabidopsis plants. (**A**) Cells of all plants exhibited well-defined structures with clear organelle contours at 24 °C; (**B**) all cells suffered varying degrees of damage; however, OE-*TaNAC6A* plants showed no significant changes after 3 days of 4 °C cold stress. Ch: chloroplast. Cw: cell wall. M: mitochondrion. Sg: starch granule. Scale bars = 2 μm.

**Figure 13 plants-14-02849-f013:**
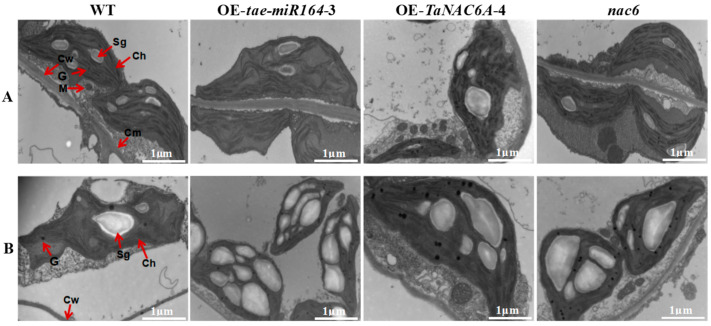
TEM micrographs of organellar morphologies undergoing changes in Arabidopsis plants. (**A**) Chloroplasts and mitochondria of all cells are normal at 24 °C; (**B**) chloroplasts and mitochondria of OE-*TaNAC6A* plants still remained normal after 3 days of 4 °C cold stress. Ch: chloroplast. Cw: cell wall. Cm: cell plasma membrane. M: mitochondrion. Sg: starch granule. G: osmiophilic body. Scale bars = 1 μm.

**Figure 14 plants-14-02849-f014:**
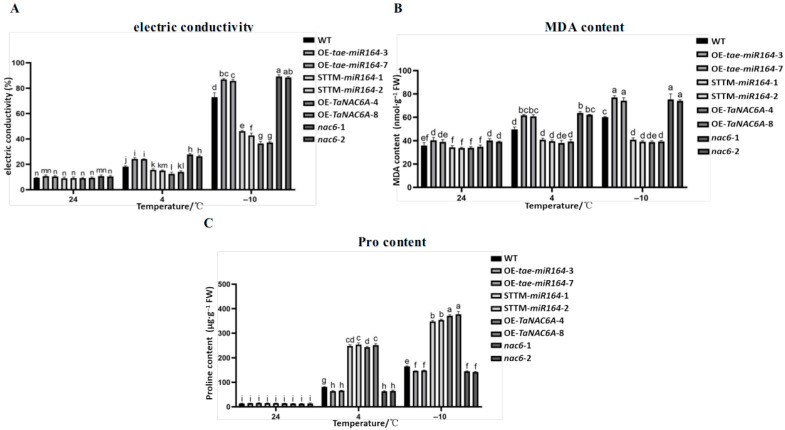
Physiological markers of cold tolerance in transgenic Arabidopsis plants. (**A**) Electric conductivity; (**B**) MDA content; (**C**) Pro content. Values represent the means ± SDs (n = 10). Different lowercase letters indicate significant differences between treatments (*p* < 0.05), determined by two-way ANOVA.

**Figure 15 plants-14-02849-f015:**
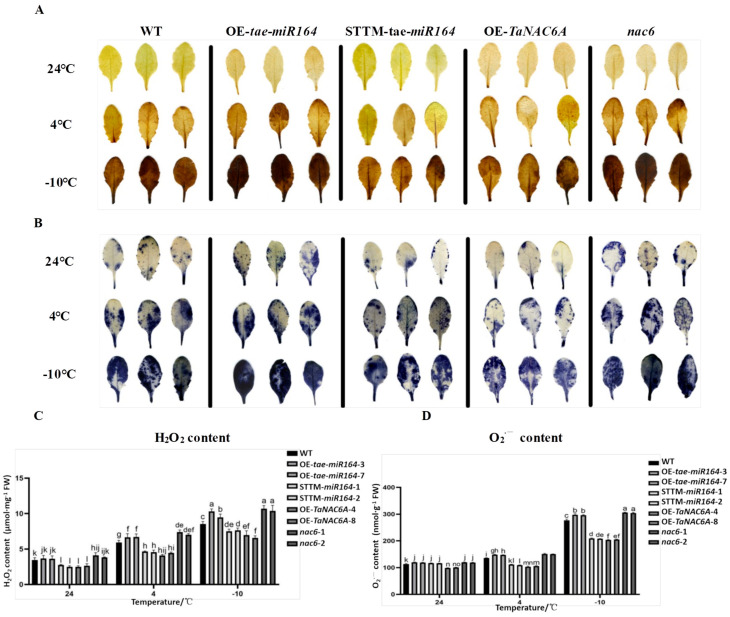
Changes in ROS levels in Arabidopsis plants under cold stress. (**A**) Leaves stained by DAB; (**B**) leaves stained by NBT; (**C**) H_2_O_2_ content; (**D**) O_2_^·−^ content. Values represent the means ± SDs (n = 10). Different lowercase letters indicate significant differences between treatments (*p* < 0.05), determined by two-way ANOVA.

**Figure 16 plants-14-02849-f016:**
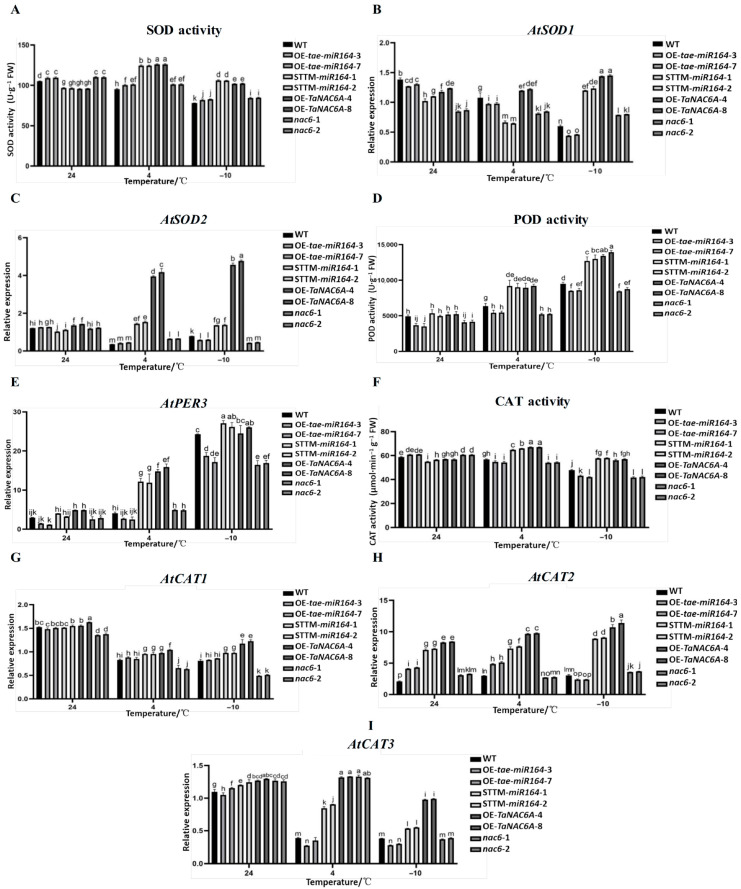
Changes in antioxidant enzyme activities and gene expression levels in Arabidopsis plants under cold stress. (**A**) SOD activity; (**B**) expression level of *AtSOD1*; (**C**) expression level of *AtSOD2*; (**D**) POD activity; (**E**) expression level of *AtPER3*; (**F**) CAT activity; (**G**) expression level of *AtCAT1*; (**H**) expression level of *AtCAT2*; (**I**) expression level of *AtCAT3*. Values represent the means ± SDs (n = 10). Different lowercase letters indicate significant differences between treatments (*p* < 0.05), determined by two-way ANOVA. The reference gene used is *AtActin*.

**Figure 17 plants-14-02849-f017:**
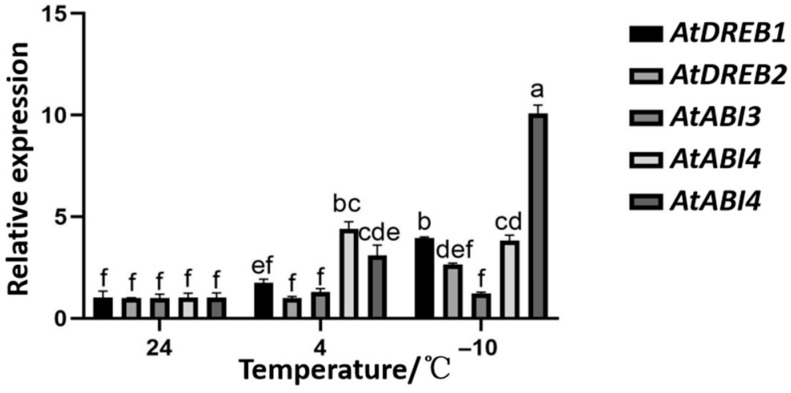
Relative expression levels of downstream target genes in Arabidopsis plants. Values represent the means ± SDs (n = 10). Different lowercase letters indicate significant differences between treatments (*p* < 0.05), determined by two-way ANOVA. The reference gene used is *AtAction*.

## Data Availability

The authors confirm that all data, tables, and figures in this manuscript are original.

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
