# Peer review of "The tae-miR164-TaNAC6A Module from Winter Wheat Could Enhance Cold Tolerance in Transgenic Arabidopsis thaliana"

_plants, 2025, doi:10.3390/plants14182849_

Round 1

Reviewer 1 Report

Comments and Suggestions for Authors

Review:  The tae-miR164-TaNAC6A Module in Winter Wheat Could Enhance Cold Tolerance of Arabidopsis thalinana

In this study the authors explore who the microRNA miR164 can regulate the expression of the TaNAC6A transcription factor to control the cold acclimation response in plants. By using a number of experiments that include transformations of tobacco, Arabidopsis and wheat plants they demonstrate that the expression of miR164 downregulates TaNAC6A and this is important because in Dn1 wheat plants miR164 is downregulated and TaNAC6A is upregulated in response to cold. The overexpression of TaNAC6A is also associated with improved cold tolerance and the overexpression of miR164 decreased cold tolerance. Plants with high TaNAC6A expression also had higher antioxidant enzyme activity and lower levels of ROS and MDA. My background is mostly looking at transcriptomic responses to stress and not gene editing experiments but the multiple lines of evidence the authors have combined seem to be strong. However, I think some details for the methods could be better explained and the conclusions in some areas of the paper are a bit beyond the scope of the evidence.

Comments

Line 35: You use response and responsive in the same sentence, a general comment after reading the paper is that I think your writing could be clearer if you avoid reusing the same words within a sentence. For example, this sentence could read “by altering cold-responsive gene expression to acclimate the plants physiology and biochemistry”.

Line 36: Maybe this should say "Frost damage occurs at critical temperatures when ice nucleation begins". Plant cells generally freeze at well below 0°C and cold stress damage can also accumulate before 0°C. I just think the wording here is a bit confusing and not in line with what you want the paper to say.

Line 44-46: I think “production” is the wrong word. ROS is a biproduct that accumulates under stress due to metabolic pathways becoming less efficient. I think “results in the accumulation of ROS” would be more appropriate.

Line 50-54: If accumulation diminished in both varieties why mention that one is sensitive and one is not? I feel like something is missing? Like the response is universal but its strength varies?

Line 55: No good link to previous paragraph.

Line 92-94: Is it possible tae-miR164 regulates any other genes?

Results 2.1: I think the text in Figure S1 and F2 is a bit small. I also think including in the main text the specific names of the cis-acting elements e.g. DRE/CRT cis-acting element, would be good.

Line 126-128: I think these conclusions are not suitable for the results and you don’t show any link to cold tolerance here and you can't confirm this is the only or main factor that regulates Dn1's response to low temperature. See also 152-153, 200-201

Figure 3 and 4: I think the methods could have a bit more of an explanation for how the GUS phenotyping works.

Lines 221-222: It looks like 80% not 40% survival in the figure and later you say 80%.

Line 283: Shouldn’t this be four plant genotypes not species?

Line 293: You never explain the OE genotypes, how large a difference was there between OE and WT for NAC6A and miR164 expression.

Lines 350-376: I don't think you need such a detailed description of the figure this could be much shorter. Overall, the results seem to be a bit too descriptive and wordy when talking about the figures.

Figure 16: Should be stand alone with the full names of genes included in the legend check other legends also.

Lines 393-397: There is no control genotype for this figure. Cold stress will stimulate the response of many other transcription factors that could also be regulating the observed response we have no evidence that this is different to the WT plants.

Lines 413-415: It is still not clear if miR164 could be regulating other wheat genes? I think we need more discussion of what this means for the results.

Lines 449-450: Again, I don't think you have the results in Figure 17 to support this statement. You don't have control data to show the response of these genes is higher than normal.

Line 484: What is the nac6 mutant I think it needs a better introduction in the results as well.

Conclusion: If you are going to put the methods last put the conclusion paragraph at the end of the discussion.

Reviewer 2 Report

Comments and Suggestions for Authors

The study investigates a wheat tae-miR164–TaNAC6A regulatory module using a combination of promoter assays, 5' RACE, transient tobacco assays, dual-luciferase, and heterologous Arabidopsis transgenics. The topic is timely and potentially useful for understanding cold tolerance; however, there are substantive issues in scope, mechanistic support, species relevance, experimental design transparency, statistics, and figure presentation. These concerns collectively preclude acceptance in the current form.

  1. Scope and title accuracy: the title and narrative imply demonstrated enhancement of cold tolerance in wheat, yet nearly all functional data are in Arabidopsis. Please revise the title and text to reflect heterologous validation in Arabidopsis, or add functional data in wheat. Also correct the species name typo in the title (page 1, “Arabidopsis thalinana”) and ensure the scope does not overstate wheat outcomes. Failure to correct such a basic error in the title is unacceptable and must be addressed prior to further consideration.
  2. Overstatement of mechanism: the Abstract claims the “molecular mechanism … was elucidated,” which is not supported by the current data that focus on target validation and phenotypic assays without downstream target binding or genome-wide transcriptional analyses. Please temper this claim or add direct mechanistic evidence such as ChIP-qPCR/ChIP-seq for TaNAC6A on antioxidant or stress-response promoters, or RNA-seq of transgenics under cold.
  1. Gene identification and homoeolog context missing (lines 109–113): TaNAC6A is presented without full wheat gene model IDs, genome coordinates, or clarification of which homoeolog (A/B/D) was cloned. In polyploid wheat, this is essential. Primer specificity against other NACs should also be documented.
  2. Arabidopsis orthology rationale unclear (lines 213–223): The inclusion of the Arabidopsis nac6 mutant assumes orthology to wheat TaNAC6A, but this is not demonstrated. A phylogenetic tree showing sequence similarity and conserved domains between TaNAC6A and AtNAC6 should be included, along with discussion of whether tae-miR164 could target Arabidopsis NACs.
  3. Promoter and dual-luciferase assays lack critical detail (Figure 3; lines 184–196): The GUS assays are qualitative only; no quantitative reporter activity is provided. For the luciferase assay, the number of biological replicates, normalization method (Firefly/Renilla), and statistical analysis are not stated. Controls with mutated binding sites are mentioned but results are not quantified in the figure.
  4. 5′ RACE validation incomplete (lines 157–162): The cleavage site mapping is described, but the frequency of cleavage events and alignment of tae-miR164 to TaNAC6A are absent. Without these, the specificity of the interaction is not fully supported.
  5. Transgenic line replication and independence (lines 203–212; lines 239–243; Figure 9): While multiple lines are mentioned, figures do not clearly specify which were used for each experiment, whether data represent multiple independent events, or if phenotypes are consistent across lines. This is critical to avoid positional effect bias.
  6. Statistical analysis insufficiently described (lines 586–590): The methods mention t-tests and ANOVA but do not map tests to specific datasets or state assumptions checked. Multiple-comparison correction and exact sample sizes per group should be reported in figure captions.
  7. ROS and physiological assays require quantitation (lines 332–339): DAB/NBT staining is shown as images without quantification. Image-based ROS quantification or colorimetric measurement should be presented, along with precise n-values for MDA, Pro, and conductivity assays.
  8. Subcellular localization needs co-localization controls (lines 134–141): Figure 2 shows nuclear localization but lacks a nuclear marker overlay. Including a DAPI or RFP-nuclear marker would strengthen this conclusion.
  9. Several captions have formatting errors (e.g., “Figure5.”), inconsistent unit spacing (“scale bars=6cm”), or unrealistic scale references for whole plants. All figures should include accurate, consistent scale bars.
  10. Terminology and nomenclature errors (line 3; line 281; line 496): Species names are misspelled (“Arabidopsis thalinana”, “Arbidopsis”, “Arabidopsisi”). Gene and miRNA nomenclature should follow standard conventions throughout text and figures.
Comments on the Quality of English Language

While the manuscript is generally understandable, the English requires editing for scientific clarity and consistency. Common issues include typographical errors in species names, inconsistent figure caption formatting, and repetitive or awkward phrasing that overstates findings. A professional language edit and basic corrections are recommended to correct terminology, improve sentence structure, and ensure consistent style.

Reviewer 3 Report

Comments and Suggestions for Authors

The manuscript titled "The tae-miR164-TaNAC6A Module in Winter Wheat Could Enhance Cold Tolerance of Arabidopsis thaliana" presents a comprehensive study on the role of the tae-miR164-TaNAC6A module in cold stress tolerance in winter wheat and Arabidopsis. The research is well-designed, employing a variety of techniques such as bioinformatics, RT-qPCR, 5'RACE, transient expression assays, and physiological analyses to elucidate the molecular mechanisms. The findings are significant, as they provide insights into the regulatory roles of miRNAs and NAC transcription factors in plant stress responses, with potential applications in molecular breeding for cold-tolerant crops.

However, the manuscript could benefit from improvements in clarity, organization, and presentation of data. Some sections require more detailed explanations, and the figures and tables need to be more consistently labeled and referenced. Additionally, the discussion could be strengthened by comparing the results more extensively with existing literature.

  1. Title Clarity:The title could be more concise. Consider revising to: "The tae-miR164-TaNAC6A Module Enhances Cold Tolerance in Winter Wheat and Arabidopsis."
  2. Abstract Structure:The abstract is lengthy and could be streamlined. Focus on key findings and implications. For example, the sentence about "molecular breeding strategies" could be more specific about potential applications.
  3. Introduction:The introduction provides a good background but could better highlight the novelty of the study. Explicitly state how this work advances existing knowledge on miR164 and NAC TFs in cold stress.
  4. Figures and Tables:Ensure all figures and tables are consistently referenced in the text. For example, Figure S1 and S2 are mentioned but not clearly integrated into the results section. The clarity of all the images is insufficient, especially the phenotype images of transgenic plants. Please replace them.

Add scale bars to all microscopy images (e.g., Figures 10, 11, 12, 13) for better interpretation.

  1. Methods Section:Provide more details on the statistical methods used, including the specific type of ANOVA and post-hoc tests applied. Clarify the sample sizes for each experiment (e.g., number of biological replicates).
  2. Results Section:The results are detailed but sometimes repetitive. For example, the expression trends of tae-miR164 and TaNAC6A in tillering nodes and leaves could be summarized more succinctly. Label subpanels in figures (e.g., Figure 1A, 1B) more clearly in the text.
  3. Discussion:Expand the discussion on how the findings compare with other studies on miR164 and NAC TFs in different plant species. For example, contrast the role of TaNAC6A with other NAC TFs like CaNAC064 or VoNAC17. Address potential limitations of the study, such as the use of heterologous expression in Arabidopsis instead of wheat.
  4. Physiological Data:The physiological data (e.g., MDA, Pro content) is well-presented but could be better contextualized. For example, explain why certain trends (e.g., Pro content) are critical for cold tolerance.
  5. Gene Expression Analysis:Clarify the rationale for selecting specific downstream genes (e.g., AIDREB1, AIDREB2) in the cold stress pathway. How do these genes interact with TaNAC6A?
  6. Technical Clarity:Define abbreviations upon first use (e.g., STTM, MRE) to improve readability for a broader audience. Ensure consistency in terminology (e.g., "hypothermia stress" vs. "cold stress").
  7. Conclusion:The conclusion is brief. Expand it to summarize key findings and their broader implications for crop improvement.
  8. Language and Grammar:Minor grammatical errors and awkward phrasing should be corrected. For example, "Lu ever found tae-miR164" should be "Lu et al. identified tae-miR164."
  9. Ethical Statement:If applicable, include a statement confirming that all experiments complied with relevant institutional guidelines.

Round 2

Reviewer 2 Report

Comments and Suggestions for Authors

The authors have addressed all the concerns raised in my review. I have no further comments and recommend acceptance of the manuscript.
